# Prepare for the Worst: Generalizing across Domain Shifts with Adversarial Batch Normalization

## Abstract

Adversarial training is the industry standard for producing models that are robust to small adversarial perturbations. However, machine learning practitioners need models that are robust to other kinds of changes that occur naturally, such as changes in the style or illumination of input images. Such changes in input distribution have been effectively modeled as shifts in the mean and variance of deep image features. We adapt adversarial training by adversarially perturbing these feature statistics, rather than image pixels, to produce models that are robust to distributional shifts. We also visualize images from adversarially crafted distributions. Our method, Adversarial Batch Normalization (AdvBN), significantly improves the performance of ResNet-50 on ImageNet-C (+8.1%), Stylized-ImageNet (+6.7%), and ImageNet-Instagram (+3.9%) over standard training practices. In addition, we demonstrate that AdvBN can also improve generalization on semantic segmentation.

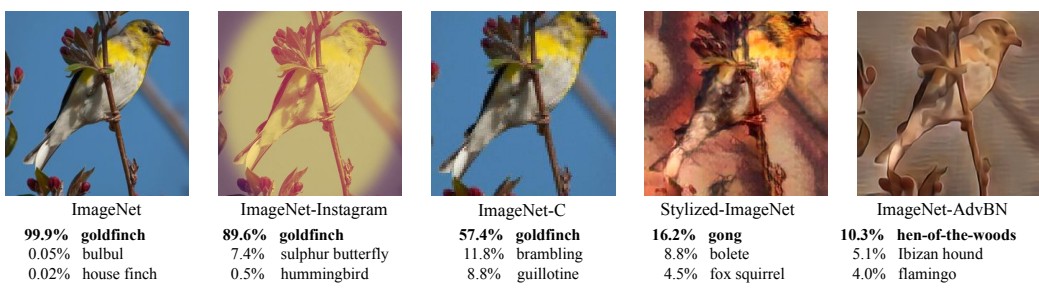

| ImageNet | ImageNet-Instagram | ImageNet-C | Stylized-ImageNet | ImageNet-AdvBN |
|---|---|---|---|---|
| **99.9% goldfinch** | **89.6% goldfinch** | **57.4% goldfinch** | **16.2% gong** | **10.3% hen-of-the-woods** |
| 0.05% bulbul | 7.4% sulphur butterfly | 11.8% brambling | 8.8% bolete | 5.1% Ibizan hound |
| 0.02% house finch | 0.5% hummingbird | 8.8% guillotine | 4.5% fox squirrel | 4.0% flamingo |

Figure 1: **Images from ImageNet variants along with classification scores by a pre-trained ResNet-50 model**. The right-most image is generated by our Adversarial Batch Normalization module. Details on how we generate this image can be found in Section 3.

## 1 Introduction

Robust optimization for neural networks has been a major focus of recent research. A mainstream approach to reducing the brittleness of classifiers is *adversarial training*, which solves a min-max optimization problem in which an adversary makes perturbations to images to degrade network performance, while the network adapts its parameters to resist degradation (Goodfellow et al., 2015; Kurakin et al., 2017; Madry et al., 2018). The result is a hardened network that is no longer brittle to small perturbations to input pixels. While adversarial training makes networks robust to adversarial perturbations, it does not address other forms of brittleness that plague vision systems. For example, shifts in image style, lighting, color mapping, and domain shifts can still severely degrade the performance of neural networks (Hendrycks & Dietterich, 2019).

We propose adapting adversarial training to make neural networks robust to changes in image style and appearance, rather than small perturbations at the pixel level. We formulate a min-max game in which an adversary chooses *adversarial feature statistics*, and network parameters are then updated to resist these changes in feature space that correspond to appearance differences of input images. This

game is played until the network is robust to a variety of changes in image space including texture, color, brightness, *etc*.

The idea of adversarial feature statistics is inspired by the observation that the mean and variance of features maps encode style information, and thus, they enable the transfer of style information from a source image to a target image through normalization (Huang & Belongie, 2017; Ulyanov et al., 2016). Unlike standard approaches that rely on feature statistics from auxiliary images to define an image style, we use adversarial optimization of feature statistics to prepare classifiers for the worst-case style that they might encounter.

We propose training with *Adversarial Batch Normalization* (AdvBN) layers. Before each gradient update, the AdvBN layers perform an adversarial feature shift by re-normalizing with the most damaging mean and variance. By using these layers in a robust optimization framework, we create networks which are resistant to any domain shift caused by feature statistics shift. An advantage of this method is that it does not require additional auxiliary data from new domains. We show that robust training with AdvBN layers hardens classifiers against changes in image appearance and style using a range of vision tasks including Stylized-ImageNet (Geirhos et al., 2019) and ImageNet-Instagram (Wu et al., 2020).

## 2 BACKGROUND

### 2.1 FEATURE NORMALIZATION

Feature normalization is an important component of modern neural networks that stabilizes training and improves model generalization. Let $f \in \mathbb{R}^{N \times C \times H \times W}$ denote feature maps output by a layer, where $N$ is the batch size, $C$ is the number of channels, and $H$ and $W$ represent the height and width of the feature maps, respectively. Different normalization methods compute the mean, $\mu$, and standard deviation, $\sigma$, over different dimensions of the feature maps. They use the derived feature statistics, often along with learned multiplicative and additive parameters, to produce normalized features, $f'$:

$$f' = \gamma \cdot \frac{f - \mu(f)}{\sigma(f)} + \beta, \tag{1}$$

where $\gamma$ and $\beta$ are learnable parameters which re-scale and shift normalized features. For example, Batch Normalization (BN) (Ioffe & Szegedy, 2015) estimates feature statistics along the $N, H, W$ dimensions. On the other hand, Instance Normalization (IN) (Ulyanov et al., 2016) computes $\mu$ and $\sigma$ for each individual sample in the batch and only normalizes across the $H$ and $W$ dimensions.

Although feature normalization was originally proposed to accelerate the training process (Bjorck et al., 2018), previous work (Huang & Belongie, 2017; Li et al., 2017) has shown that feature statistics effectively capture information concerning the appearance of images. Motivated by this observation, we impose uncertainty on these statistics during training in order to obtain models that are less sensitive to non-semantic characteristics, thus generalizing to images with different appearances.

### 2.2 ADVERSARIAL TRAINING

Untargeted adversarial examples are generated by maximizing classification loss with respect to the input. One popular method, projected gradient descent (PGD), involves performing gradient ascent in the signed gradient direction and projecting the perturbation in order to enforce an $\ell_\infty$-norm constraint (Madry et al., 2018). Adversarial training aims to solve the saddlepoint optimization problem,

$$\min_\theta \mathbb{E}_{(X,y) \sim \mathcal{D}} \left[ \max_{\|\delta\|_p < \epsilon} \mathcal{L}(g_\theta(X + \delta), y) \right], \tag{2}$$

where $g_\theta$ is a model with parameter vector $\theta$, $X, y$ is a clean input and the corresponding label drawn from distribution $\mathcal{D}$, and $\mathcal{L}$ denotes cross-entropy loss. Adversarial training solves this problem by iteratively sampling a batch of data, perturbing the batch adversarially, and performing a parameter update on the new adversarial batch (Madry et al., 2018). We harness adversarial training in order to create models robust to distributional shifts rather than the standard pixel-wise adversarial attacks.

# 3 ADVERSARIAL BATCH NORMALIZATION

We propose *Adversarial Batch Normalization* (AdvBN), a module that adversarially perturbs deep feature distributions such that the features confuse CNN classifiers. We iteratively compute adversarial directions in feature space by taking PGD steps on batch statistics. In the next section, we will train on these perturbed feature distributions in order to produce models robust to domain shifts.

Consider a pre-trained classification network, $g$, with $L$ layers. We divide $g$ into two parts, $g^{1,l}$ and $g^{l+1,L}$, where $g^{m,n}$ denotes layers $m$ through $n$. Now, consider a batch of data, $x$, with corresponding labels, $y$. Formally, the AdvBN module is defined by

$$\text{BN}_{\text{adv}}^{\delta}(x; g, l, y) = \delta_{\sigma}' \cdot (f - \mu(f)) + \delta_{\mu}' \cdot \mu(f), \text{ where } f = g^{1,l}(x),$$

$$(\delta_{\mu}', \delta_{\sigma}') = \arg\max_{(\delta_{\mu}, \delta_{\sigma})} \mathcal{L}\left[g^{l+1,L}\left(\delta_{\sigma} \cdot (f - \mu(f)) + \delta_{\mu} \cdot \mu(f)\right), y\right], \tag{3}$$

$$\text{subject to } \|\delta_{\mu} - 1\|_{\infty} \leq \epsilon, \|\delta_{\sigma} - 1\|_{\infty} \leq \epsilon,$$

where the maximization problem is solved with projected gradient descent. Simply put, the AdvBN module is a PGD attack on batch norm statistics which can be inserted inside a network. Note that $\delta_{\mu}, \delta_{\sigma}$ are vectors with length equal to the number of channels in the output of layer $l$, and we multiply by them entry-wise, one scalar entry per channel, similarly to Batch Normalization. Additionally, note that this module acts on a per-batch basis so that features corresponding to an individual image may be perturbed differently depending on the batch the image is in.

We formulate the perturbation by first subtracting out the mean so that $\delta_{\mu} \cdot \mu(f)$ is the new mean of the adversarial features, and $\delta_{\mu}$ directly controls the new mean. We choose $\delta_{\mu} \cdot \mu(f)$, rather than simply $\delta_{\mu}$, to represent the new mean of the perturbed features so that $\ell_{\infty}$ bounds and steps size do not need to depend on the mean of $f$.

**Visualizing feature shifts** To verify our assumption that adversarially perturbing feature statistics corresponds to transforming the distribution in image space, we visualize the effects of AdvBN. We adopt the VGG-19 based autoencoder from Huang & Belongie (2017). At the bottleneck of the autoencoder, we plug in an AdvBN module We visualize perturbations by feeding the AdvBN outputs to the decoder.

In Figure 2, images crafted through this procedure are visibly different from the originals; semantic content in the original images is preserved, but the new images exhibit differences in color, texture, and edges. We draw two major conclusions from these visualizations which highlight the adversarial properties of these domains. The first one concerns textures: according to Geirhos et al. (2019), CNNs rely heavily on image textures for classification. Images from the adversarial domain, on the other hand, have inconsistent textures across samples. For example, the furry texture of a dog is smoothed in column 2, and the stripes disappear from a zebra in column 4, whereas visible textures appear in columns 6 and 8. The second conclusion pertains to color. Results in Zhang et al. (2016) suggest that colors serve as important information for CNNs. In the adversarial domain, we find

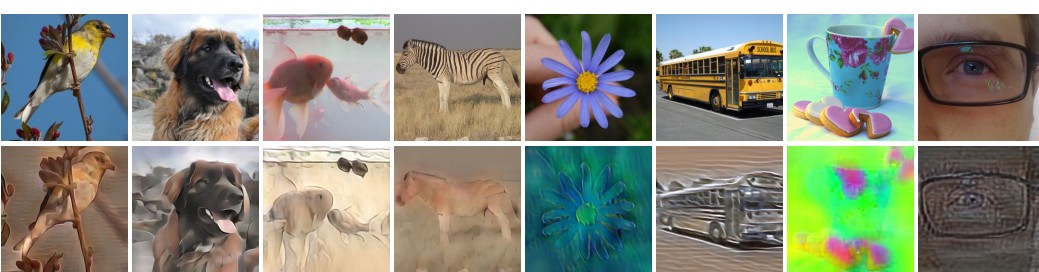

Figure 2: **Examples of perturbed ImageNet images generated by AdvBN with a decoder**. The first row contains the original versions from the ImageNet validation set.

suppressed colors (columns 1, 3) and unnatural hue (columns 5, 7). See Appendix A.3 for additional example images generated by this procedure. Figure 3 illustrates how the appearance of reconstructed images shifts as adversarial perturbations to feature statistics become larger.

We use this visualization technique to process the entire ImageNet validation set and denote it as ImageNet-AdvBN in Figure 1. By evaluating different methods on this dataset, we observe that performance on ImageNet-AdvBN is consistently degraded, which validates the adversarial property of features generated by AdvBN. Experiments concerning performance on ImageNet-AdvBN are listed in Appendix A.2.

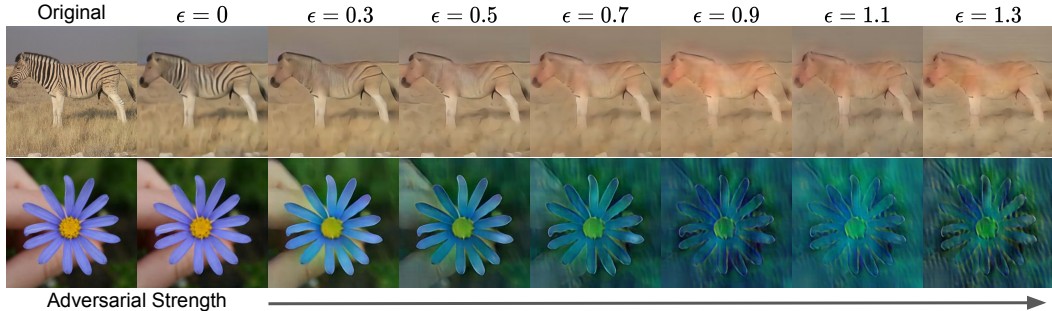

Figure 3: **The effect of adversarial strength on visualized examples**. $\epsilon = 0$ corresponds to images reconstructed by our autoencoder without AdvBN.

## 4 TRAINING WITH ADVERSARIAL BATCH NORMALIZATION

In this section, we use the proposed AdvBN module to train networks on the perturbed features. The goal is to produce networks that generalize well to unseen domains while maintain performance on training data distribution, all without having to obtain auxiliary data from new domains.

We start with a pre-trained model, $g = g^{l+1,L} \circ g^{1,l}$, and we fine-tune the subnetwork, $g^{l+1,L}$, on clean and adversarial features simultaneously. To this end, we solve the following min-max problem,

$$\min_{\theta} \mathbb{E}_{(x,y)\sim\mathcal{D}} \left[ \max_{\delta} \mathcal{L}(g_{\theta}^{l+1,L} \circ \text{BN}_{\text{adv}}^{\delta} \circ g^{1,l}(x), y) + \mathcal{L}(g_{\theta}^{l+1,L} \circ g^{1,l}(x), y) \right], \tag{4}$$

where $\mathcal{L}$ denotes cross-entropy loss, and $\mathcal{D}$ is the distribution of batches of size $n$. In order to maintain the network's performance on natural images, we adopt a similar approach to Xie et al. (2020) by using auxiliary batch normalization in $g^{l+1,L}$ for adversarial features; we use the original BNs when propagating clean features, and we use auxiliary ones for adversarial features. See Algorithm 1 for a detailed description of our method.

Since we start with pre-trained models, we only need to fine-tune for 20 epochs, yielding improved robustness with little additional computation. Moreover, we only modify the parameters of later layers, so we do not need to backpropagate through the first half of the network. See Appendix B for an analysis of training budget using our method. In the following section, we measure the performance, on several datasets, of our model fine-tuned using adversarial training with AdvBN.

## 5 EXPERIMENTS

### 5.1 IMPLEMENTATION

Our method begins with a torchvision ImageNet pre-trained ResNet-50 (He et al., 2016). We insert the AdvBN module at the end of the 2nd convolutional stage. The model is fine-tuned following Algorithm 1 for 20 epochs. The learning rate starts at 0.001 and decreases by a factor of 10 after 10 epochs with a batch size of 256. We use SGD with momentum 0.9 and weight decay coefficient $10^{-4}$. We augment inputs with a fixed AutoAugment (Cubuk et al., 2019) policy. Adversarial parameters are $\tau = 0.2$ and $\epsilon = 1.1$ with 6 repeats.

---

**Algorithm 1:** Training with Adversarial Batch Normalization

---

**Input:** Training data, pretrained network $g = g_\theta^{l+1,L} \circ g^{1,l}$, PGD bound $\epsilon$, and PGD step size $\tau$
**Result:** Updated network parameters, $\theta$, of subnetwork $g_\theta^{l+1,L}$
**for** *each training step* **do**
  Sample mini-batch $x$ with label $y$;
  Obtain feature map $f = g^{1,l}(x)$;
  Initialize perturbation: $\delta = (\delta_\mu, \delta_\sigma)$;
  Let $f_{adv} = f$;
  **for** *adversarial step = 1, ..., m* **do**
    $f_{adv} \leftarrow \delta_\sigma \cdot (f - \mu(f)) + \delta_\mu \cdot \mu(f)$;
    Update $\delta$: $\delta \leftarrow \delta + \tau \cdot sign(\nabla_\delta \mathcal{L}(g_\theta^{l+1,L}(f_{adv}), y))$;
    $\delta \leftarrow$ clip $(\delta, 1 - \epsilon, 1 + \epsilon)$;
  **end**
  $f_{adv} \leftarrow \delta_\sigma \cdot (f - \mu(f)) + \delta_\mu \cdot \mu(f)$;
  Minimize the total loss w.r.t. network parameter:
    $\theta \leftarrow \underset{\theta}{\arg\min} \mathcal{L}(g_\theta^{l+1,L}(f_{adv}), y) + \mathcal{L}(g_\theta^{l+1,L}(f), y)$;
**end**
**return** $\theta$

---

## 5.2 GENERALIZATION TO IMAGENET VARIANTS

**Datasets.** We compare our method to other methods designed to produce classification networks which generalize better. The datasets we consider are variants of ImageNet (Deng et al., 2009):

- **ImageNet-C** (Hendrycks & Dietterich, 2019) contains distorted images with 15 categories of common image corruption applied, each with 5 levels of severity. Performance on this dataset is measured by mean Corruption Error (mCE), the average classification error over all 75 combinations of corruption type and severity level, weighted by their difficulty.

- **ImageNet-Instagram** (Wu et al., 2020) is composed of ImageNet images filtered with a total of 20 different Instagram filters. This dataset contains 20 versions of each ImageNet image, each with a different filter applied.

- **Stylized-ImageNet** (Geirhos et al., 2019) consists of images from the ImageNet dataset, each stylized using AdaIN (Huang & Belongie, 2017) with a randomly selected painting. Textures and colors of images in this dataset differ heavily from the originals.

**Models.** Our baseline model is the publicly available torchvision ResNet-50 pre-trained on ImageNet, denoted as "Standard" in Table 1. All models we compare to, aside from SIN (Geirhos et al., 2019), are not trained on any of the ImageNet variants that are used for evaluation. The PGD model is adversarially trained with the PGD attack on inputs and is provided by Engstrom et al. (2019). MoEx (Li et al., 2020) changes feature moments inside networks but not in an adversarial manner. IBN-Net (Pan et al., 2018) improves the generalization of networks by combining batch normalization and instance normalization. AugMix (Hendrycks et al., 2020) is a data augmentation method that solves the distributional mismatch between training and testing data and increases classification robustness. SIN is a network trained on both Stylized ImageNet and ImageNet. We do not measure the accuracy of SIN on Stylized-ImageNet since it acquires knowledge of the target domain during training. Note that all models we use in our comparisons are the original versions released by the authors of the original work.

**Results.** As shown in Table 1, the performance of baseline model significantly degrades on all three ImageNet variants, highlighting the brittleness of standard classification model when tested on novel distributions. Fine-tuning with AdvBN, on the other hand, substantially improves the performance of the standard ResNet-50 model. In particular, we achieve an 8.1% accuracy gain on ImageNet-C through fine-tuning with AdvBN. On Stylized-ImageNet and ImageNet-Instagram, our model also achieves the best performance among all methods with which we compare. The consistent performance boost across all three benchmarks demonstrates that AdvBN can effectively enhance robustness against various distributional shifts. Note that this AdvBN model has additional auxiliary

Table 1: **Performance of AdvBN and alternative methods on ImageNet variants**.

| Model | ImageNet-C mCE ↓ | ImageNet-Instagram top1/ top5 acc. ↑ | Stylized-ImageNet top1/ top5 acc.↑ |
|---|---|---|---|
| Standard | 76.7 | 67.2/ 87.6 | 7.4/ 16.4 |
| PGD | 85.0 | 49.0/ 71.4 | 12.5/ 23.9 |
| MoEx | 74.8 | 70.0/ 89.4 | 5.0/ 12.0 |
| IBN-Net | 70.3 | 69.6/ 89.3 | 10.7/ 22.2 |
| Augmix | **68.4** | 70.4/ 89.4 | 11.2/ 23.1 |
| SIN | 69.3 | 66.9/ 87.4 | –/ – |
| AdvBN | 68.6 | **71.1/ 89.5** | **14.1/ 26.9** |

BN layers, so its performance on the original ImageNet is well maintained, as will be shown in Table 4 in the next subsection. Appendix C gives details of inference with auxiliary BN layers. Besides ResNet-50, we also applied AdvBN to fine-tune other model architectures. From Table 2, we see that AdvBN improves the performance of both DenseNet and EfficientNet architectures.

Table 2: **Applying AdvBN to other architectures**.

| Architecture | ImageNet-C mCE. ↓ | ImageNet-Ins. top1 acc. ↑ | ImageNet-Styl. top1 acc. ↑ |
|---|---|---|---|
| DenseNet-121 | 73.4 | 66.6 | 7.9 |
| + AdvBN | **70.4** | **69.3** | **15.5** |
| EfficientNet-B0 | 72.1 | 69.7 | 12.5 |
| + AdvBN | **68.7** | **71.3** | **15.7** |

Table 3: **Performance of AdvBN and an alternative method: AdvProp, on EfficientNet-B0**.

| Dataset | AdvProp | AdvBN |
|---|---|---|
| ImageNet-C | **66.2** | 68.7 |
| ImageNet-Ins. | 70.6 | **71.3** |
| ImageNet-Styl. | 14.6 | **15.7** |

With our results on EfficientNet, we are able to compare to another alternative method, AdvProp(Xie et al., 2020), which also adopts adversarial training framework for improving the performance of neural networks. Since no official ResNet-50 model of this method is available, we list the comparison results in Table 3 separately with a EfficientNet-B0 architecture.

## 5.3 ABLATION STUDY

**Where should the AdvBN module be placed within a network?** The proposed AdvBN module can be inserted after any layer in a network. In this part, we try AdvBN with different positions, `conv3_4` and `conv4_6`. From the results in Table 4, we observe that `conv4_6` yields the worst performance among all three ImageNet variants, indicating that using AdvBN at such deep layer is not as helpful as at shallower layers. We hypothesize two possible explanations for this phenomenon: (1) there are fewer trainable parameters when only very deep layers are fine-tuned; (2) features are more abstract in deeper layers, and perturbing these high-level features can lead to extremely chaotic feature representations that are harmful for classification.

Table 4: **Ablation studies**.

| Model | ImageNet top1 acc. ↑ | ImageNet-C mCE. ↓ | ImageNet-Ins. top1 acc. ↑ | ImageNet-Styl. top1 acc. ↑ |
|---|---|---|---|---|
| Standard | 76.1 | 76.7 | 67.2 | 7.4 |
| $l = $ `conv2_3` | 76.5 | 68.6 | 71.1 | 14.1 |
| $l = $ `conv3_4` | 76.0 | 70.0 | 70.2 | 19.5 |
| $l = $ `conv4_6` | 75.3 | 75.0 | 68.5 | 11.0 |
| $\epsilon = 0.5$ | 76.5 | 69.4 | 70.8 | 13.3 |
| $\epsilon = 0.7$ | 76.4 | 69.0 | 70.9 | 13.5 |
| $\epsilon = 0.9$ | 76.6 | 69.0 | 71.0 | 13.8 |
| $\epsilon = 1.1$ | 76.5 | 68.6 | 71.1 | 14.1 |
| $\epsilon = 1.3$ | 76.4 | 68.3 | 70.7 | 13.2 |
| $\epsilon = 1.5$ | 76.2 | 68.8 | 70.4 | 13.1 |
| AutoAugment* | 76.4 | 72.1 | 70.1 | 8.2 |

**Adversarial strength.** The strength of the adversarial attack in the adversarial training framework has a major impact on model performance (Madry et al., 2018). We test a range of PGD parameters to demonstrate how the strength of AdvBN affects model performance. We measure strength by the perturbation bound $\epsilon$, where we fix $\tau$ to be 0.2 for all settings, and change the number of repeats for

different bounds. The number of repeats $n$ for each $\epsilon$ is set to be $n = [\epsilon/0.2] + 1$. Results concerning the impact of adversarial strength are listed in Table 4.

**Data augmentation.** AdvBN performing in feature space can easily be combined with input space data augmentation. To determine what portion of the improvements we observed can be credited to AdvBN, we fine-tune a pre-trained ResNet-50 following the same routine as before but without the AdvBN module, adopting the same fixed AutoAugment policy along with all other hyperparameters. This method is denoted by AutoAugment* in Table 4. We see that fine-tuning with AutoAugment alone does not result in nearly as much improvement as the combined method on all datasets we consider; even performance on the original ImageNet dataset benefits from the AdvBN module.

## 5.4 FEATURE DIVERGENCE ANALYSIS

We compare the features extracted by our network to those of a standard ResNet-50 trained on ImageNet. Following Pan et al. (2018), we model features from each channel using a normal distribution with the same mean and standard deviation, and we compute the symmetric KL divergence between the corresponding distributions on the two datasets ($A$ and $B$). For two sets of deep features, $F_A$ and $F_B$, each with $C$ channels, the divergence $D(F_A||F_B)$ is computed using the formula,

$$D(F_A||F_B) = \frac{1}{C}\sum_{i=1}^{C}(KL(F_A^i||F_B^i) + KL(F_B^i||F_A^i)),$$

$$KL(F_A^i||F_B^i) = \log\frac{\sigma_B^i}{\sigma_A^i} + \frac{\sigma_A^{i^2} + (\mu_A^i - \mu_B^i)^2}{2\sigma_B^{i^2}} - \frac{1}{2}, \quad (5)$$

where $F^i$ denotes the features of $i$-th channel with mean $\mu^i$ and standard deviation $\sigma^i$.
In Figure 4, we compare the baseline model with our own on two pairs of datasets in the fine-tuned layers. Since ImageNet-Instagram contains 20 filter versions, we use the "Toaster" filter found in (Wu et al., 2020) to cause the sharpest drop in classification performance. We find that the feature

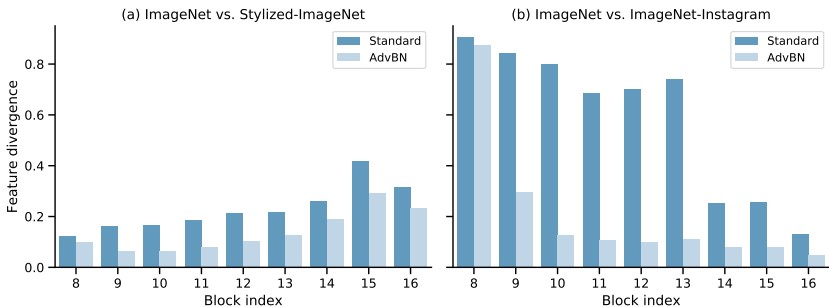

Figure 4: **Feature divergence between pairs of datasets using features extracted by AdvBN and a standard ResNet50**.

divergence in our network trained with AdvBN is substantially smaller on all layers in the fine-tuned subnetwork. The small divergence between feature representations explains the effectiveness of AdvBN from a different angle and explains why our model generalizes well across datasets.

## 5.5 GENERALIZATION ON SEMANTIC SEGMENTATION

We now evaluate AdvBN in the context of semantic segmentation. The segmentation model we use is a ResNet-50 based network with dilated convolutions (Yu & Koltun, 2016). We present generalization results under two scenarios: the first is from Cityscapes (Cordts et al., 2016) to GTA5 (Richter et al., 2016), where both datasets contain similar urban road scene imagesand have compatible label categories; the second is generalizing across traffic situations under different weather/illumination/season conditions with the Synthia video sequences dataset (Ros et al., 2016). Our baseline models for both scenarios are trained following the training protocol in Pan et al. (2018).

We train a baseline model on Cityscapes, which contains 2975 for training, and test the generalization performance of the model on GTA5 with 6382 images for validation. We apply AdvBN by plugging it

after layer `conv2_3` of baseline model, and fine-tuned on Cityscapes for 20 epochs, with adversarial training parameters $\tau = 0.15$, $\epsilon = 0.4$, and 4 repeats.

In table 5, we observe a performance gain of 8.2% in mean IoU on the GTA5 dataset compared to the baseline model. The pixel accuracy also improves by 11.3%. Numbers on the left side of arrows denote performance on Cityscapes, and numbers on the right side of arrows denote performance on the GTA5 dataset.

Table 5: **Cityscapes → GTA5**.

|  | mean IoU (%) | Pixel Acc. (%) |
|---|---|---|
| Baseline | 72.0 → 30.6 | 95.3 → 68.5 |
| AdvBN | 71.5 → 38.8 | 95.2 → 79.8 |

For Synthia dataset, we use the left-front view images of each sub-dataset by randomly selecting 900 images for training and 500 for validation. We consider two different road scene: "Highway" and "New York-like City", each one with 5 different domain shifted variants: "dawn", "fog", "night", "spring" and "winter". We obtain our baseline models and AdvBN fine-tuned models following the same hyperparameter settings as for Cityscapes.

Table 6: **Semantic segmentation results (mean IOU) on Synthia dataset**.

|  |  | New York-like City | | | | |  |  | Highway | | | | |
|---|---|---|---|---|---|---|---|---|---|---|---|---|---|
|  |  | Dawn | Fog | Night | Spring | Winter |  |  | Dawn | Fog | Night | Spring | Winter |
|  | baseline | 32.6 | 29.0 | 25.4 | 24.2 | 24.8 |  | baseline | 18.6 | 21.0 | 16.9 | 21.6 | 15.3 |
| Highway/Dawn | AdvBN | **33.7** | **30.7** | **28.4** | **28.5** | **27.5** | NY.Like C./ Spring | AdvBN | **21.6** | **24.2** | **22.2** | **27.2** | **19.8** |

In table 6, we compare the mean IOU of baseline models and AdvBN fine-tuned models. We train two models on "Highway/Dawn" and "New York-like City/Spring" datasets separately and test them on the opposite road scene with different weather conditions. We can see AdvBN consistently improve the performance on each train-test pairs.

## 6 RELATED WORK

**Adversarial training** Adversarial training and its variants (Goldblum et al., 2020; Madry et al., 2018; Shafahi et al., 2019) have been widely studied for producing models that are robust to adversarial examples (Moosavi-Dezfooli et al., 2016; Szegedy et al., 2014). Recent work considers adversarial training as data augmentation (Tsipras et al., 2019). Xie et al. (2020) finds that deep features corresponding to adversarial examples have different mean and standard deviation than those corresponding to natural images. This work takes advantage of the distributional discrepancy to improve performance on non-adversarial data. Our work also adopts the adversarial training framework to make models robust against other kinds of perturbations, but instead of crafting adversarial examples in image space, we craft adversarial feature distributions by perturbing feature statistics.

**Robustness to distributional shifts** While extensive effort has been made to improve the robustness of classifiers to adversarial examples, there are other kinds of robustness that deep neural networks must address in order for them to be reliable. Corrupted images and new domains pose major challenges to networks with standard training (Geirhos et al., 2019; Hendrycks & Dietterich, 2019; Wu et al., 2020). Performance degradation on these images can be attributed to shifts in data distributions (Gilmer et al., 2018). In order to produce networks which generalize well, one common practice is to perform data augmentation (Cubuk et al., 2019; Hendrycks et al., 2020; Yun et al., 2019). However, the benefits of data augmentation are largely limited by the types of augmentations used during training (Geirhos et al., 2018). Feature space augmentation (Li et al., 2020) replaces feature statistics corresponding to one sample with ones corresponding to another sample. Our work can be also considered as feature space augmentation, we instead consider a worst-case scenario in the context of feature space distributional shifts by adopting the adversarial training framework.

## 7 CONCLUSION

Our work studies how adversarially perturbing feature statistics simulates distributional shift in image data. We find that adversarial fine-tuning on features perturbed in this way improves robustness to data stylization and corruption without ever training on auxiliary data. Training with Adversarial Batch Normalization (AdvBN) is computationally cheap and can quickly make pre-trained models less brittle. We fine-tune a ResNet-50 with our algorithm and surpass the performance of state-of-the-art methods on both ImageNet-Instagram and Stylized-ImageNet. Adversarial feature statistics are a promising direction for creating models that generalize well to a variety of domains.

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

# A    IMAGENET-ADVBN EXPERIMENTS

## A.1    CREATION OF THE IMAGENET-ADVBN DATASET

We process the entire ImageNet validation set using the visualization technique introduced in Section 3. We consider two encoder architectures: one is the VGG-19 encoder we use for visualization, another consists of layers of a ResNet-50 up to `conv2_3`. Both encoders are paired with the same decoder architecture from Huang & Belongie (2017). The resulting datasets, denoted by ImageNet-AdvBN-VGG and ImageNet-AdvBN-ResNet respectively, contain 50000 images each. The data we synthesize for testing other models is generated using these autoencoders that contain the AdvBN module but on ImageNet validation data. AdvBN is conducted with 6 steps, stepsize = 0.20, $\epsilon = 1.1$, and a batchsize of 32. We do not shuffle the ImageNet validation data when generating these batches.

## A.2    CLASSIFICATION ON IMAGENET-ADVBN

Table 7: **Classification performance on ImageNet and ImageNet-AdvBN**.

| Method | ImageNet top1 acc. ↑ | Im-Adv-VGG top1/ top5 acc. ↑ | VGG Reconstructed top1/ top5 acc. ↑ | Im-Adv-ResNet top1/ top5 acc. ↑ | ResNet Reconstructed top1/ top5 acc. ↑ |
|---|---|---|---|---|---|
| Standard | 76.1 | 1.6/ 4.7 | 45.8/ 70.6 | 0.4/ 1.3 | 65.7/ 86.9 |
| PGD | 62.4 | 15.4/ 30.2 | 54.7/ 77.8 | 1.0/ 2.2 | 61.2/ 83.1 |
| MoEx | 79.1 | 1.0/ 2.9 | 40.2/ 63.8 | 0.3/ 1.1 | 65.7/ 86.8 |
| IBN-Net | 77.2 | 2.3/ 6.2 | 49.2/ 73.9 | 0.6/ 1.9 | 69.4/ 89.4 |
| AugMix | 77.6 | 3.9/ 9.9 | 53.5/ 77.0 | 1.0/ 2.7 | 71.9/ 90.7 |
| SIN | 74.6 | 10.9/ 24.4 | 51.7/ 75.8 | 1.7/ 4.3 | 67.2/ 87.9 |
| Ours | 76.5 | 6.3/ 15.6 | 54.6/ 78.3 | 2.9/ 6.4 | 66.4/ 87.3 |

Table 7 shows the classification performance of various models on the two ImageNet-AdvBN variants, denoted as Im-Adv-VGG and Im-Adv-ResNet respectively. We also test these models on ImageNet images that are reconstructed using our autoencoders, denoted as VGG Reconstructed and ResNet Reconstructed, for each autoencoder. The performance gap between ImageNet-AdvBN and Reconstructed ImageNet indicates that the degradation on ImageNet-AdvBN is not solely caused by the reconstruction loss due to the autoencoders we use.

## A.3    ADDITIONAL EXAMPLE IMAGES

We include more images from ImageNet-AdvBN-VGG in this section. Example images in Figure 5 are randomly chosen. We do not include the ImageNet-AdvBN-ResNet, because the resulting images are mostly in extreme contrast with small textures that are hard to observe. It is possible that features output from ResNet based encoders are more sensitive to AdvBN perturbations; another explanation is that the features we extract from ResNet-50 are relatively shallow features compared to their VGG counterparts.

# B    RUNTIME ANALYSIS

## B.1    RUNTIME OF TRAINING WITH ADVBN

We evaluate the training time of our method on a workstation with 4 GeForce RTX 2080 Ti GPUs. We use the default settings for AdvBN on ResNet-50: an AdvBN module after the `conv2_3` layer, a fixed AutoAugment policy, and 20 epochs of fine-tuning with 6-step PGD inside the AdvBN module. Fine-tuning is conducted on the ImageNet training set, containing 1.3 million images. Training in this setting takes approximately 40 hours with batchsize set to 256 .

## B.2    COMPARISON WITH OTHER METHODS CONCERNING TRAINING BUDGET

We use the same infrastructure above to evaluate the training time of other ResNet-50 models that appear in Table 1. Training code for all other methods are obtained from official repository. For all methods, we use a batch size of 128, because augmix(Hendrycks et al., 2020) cannot run with

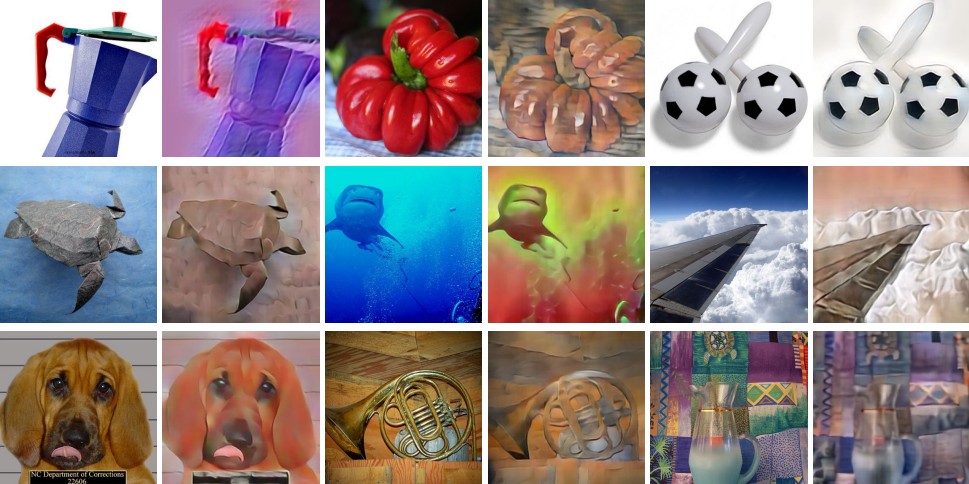

Figure 5: More example images. For each pair of adjacent columns, original versions are on the left, ImageNet-AdvBN-VGG is on the right.

a batch size of 256 on our workstation due to limited GPU memory. The number of processes in the dataloader is set to be 16. The speed values in Table 8 are averaged over 100 iterations. The estimated training duration is calculated by multiplying the speed and corresponding total number of iterations. Time spent on evaluation after each epoch is not considered in this estimation.

Table 8: **Runtime Analysis**.

| Model | Speed (seconds/ iter.) | Epochs | Estimated training duration (hours) |
|---|---|---|---|
| Standard | 0.17 | 90 | 43 |
| PGD | 2.95 | 90 | 738 |
| MoEx | 0.19 | 300 | 168 |
| IBN-Net | 0.22 | 100 | 61 |
| AugMix | 0.67 | 180 | 335 |
| SIN | 0.34 | 45 | 43 |
| AdvBN | 0.83 | 20 | 46 |

Training with AdvBN takes a long time per iteration because each iteration contains 6 PGD steps. The SIN (Geirhos et al., 2019) runtime speed is estimated based on standard training, since the model architectures and training procedures of these two methods are the same, except that SIN is trained for half the epochs but twice the data and thus twice the number of iterations per epoch. Training SIN requires additional access to Stylized-ImageNet as training data, which takes 134GB disk space; the time for generating the Stylized-ImageNet dataset and the extra storage cost are not considered in Table 8.

## C  INFERENCE USING MODELS TRAINED WITH ADVBN

Models containing Batchnorm layers will have two set of BN statistics in deeper layers that have been fine-tuned by AdvBN because we use auxiliary BNs introduced by (Xie et al., 2020) for propagating adversarial features crafted by the AdvBN module. During evaluation, we can choose either of the BN statistics to normalize features. The results we report in previous sections with regard to ImageNet, ImageNet-C and ImageNet-Instagram are obtained by using BN statistics corresponding to original features. We only use auxiliary BNs, which keep the batch statistics of adversarial features, to test performance on Stylized-ImageNet in Table 1. We also use auxiliary BNs for evaluating performances on ImageNet-AdvBN and Reconstructed ImageNet in Table 7.

# D    DETAILED RESULTS ON IMAGENET-C

In this section we provide a detailed version of the results shown in Table 1 concerning the ImageNet-C dataset, which technically contains a total of 75 variants of the ImageNet dataset. The 75 variants fall into 15 categories of corruptions, each category presents 5 gradually increasing degrees of severity, where "degree=1" denotes the lowest degree of severity. From Table 9, we can see that AdvBN is

Table 9: **Detailed results on ImageNet**.

| Network | Clean | Noise | | | Blur | | | | Weather | | | | Digital | | | | mCE |
| | | Gauss. | Shot | Impulse | Defocus | Glass | Motion | Zoom | Snow | Frost | Fog | Bright | Contrast | Elastic | Pixel | JPEG | |
| --- | --- | --- | --- | --- | --- | --- | --- | --- | --- | --- | --- | --- | --- | --- | --- | --- | --- |
| Standard | 23.9 | 80 | 82 | 83 | 75 | 89 | 78 | **80** | 78 | 75 | 66 | 57 | 71 | 85 | 77 | 77 | 76.7 |
| AdvBN | **23.5** | 65 | 64 | 65 | **70** | 84 | 75 | 82 | **71** | 70 | 58 | 52 | 49 | 85 | 70 | 71 | 68.6 |

| Model | Corruption | Degree | | | | | Corruption | Degree | | | | | Corruption | Degree | | | | | Corruption | Degree | | | | |
| | | 1 | 2 | 3 | 4 | 5 | | 1 | 2 | 3 | 4 | 5 | | 1 | 2 | 3 | 4 | 5 | | 1 | 2 | 3 | 4 | 5 |
| --- | --- | --- | --- | --- | --- | --- | --- | --- | --- | --- | --- | --- | --- | --- | --- | --- | --- | --- | --- | --- | --- | --- | --- | --- |
| AdvBN | Blur-Defocus | 38 | 44 | 56 | 69 | 79 | Blur-Glass | 39 | 52 | 80 | 86 | 90 | Blur-Motion | 33 | 42 | 59 | 76 | 85 | Blur-Zoom | 48 | 59 | 66 | 73 | 79 |
| AugMix | | 35 | 40 | 50 | 62 | 74 | | 39 | 50 | 74 | 78 | 84 | | 28 | 33 | 42 | 58 | 71 | | 38 | 45 | 49 | 57 | 65 |
| AdvBN | Weather-Snow | 42 | 62 | 59 | 69 | 75 | Weather-Frost | 36 | 52 | 63 | 65 | 71 | Weather-Fog | 33 | 37 | 44 | 52 | 69 | Weather-Bright | 26 | 27 | 28 | 31 | 35 |
| AugMix | | 39 | 59 | 57 | 69 | 77 | | 35 | 50 | 62 | 64 | 71 | | 37 | 42 | 52 | 58 | 75 | | 25 | 26 | 29 | 33 | 40 |
| AdvBN | Digital-Contrast | 29 | 30 | 34 | 46 | 68 | Digital-Elastic | 32 | 56 | 44 | 58 | 84 | Digital-Pixel | 32 | 33 | 48 | 65 | 74 | Digital-JPEG | 34 | 37 | 39 | 47 | 58 |
| AugMix | | 29 | 33 | 39 | 59 | 85 | | 31 | 53 | 37 | 48 | 71 | | 30 | 32 | 41 | 53 | 60 | | 32 | 35 | 37 | 43 | 52 |
| AdvBN | Noise-Gauss. | 35 | 42 | 54 | 69 | 86 | Noise-Shot | 35 | 43 | 54 | 71 | 82 | Noise-Impulse | 40 | 48 | 55 | 71 | 85 | | | | | | |
| AugMix | | 32 | 40 | 55 | 76 | 94 | | 33 | 42 | 55 | 77 | 88 | | 36 | 46 | 56 | 79 | 95 | | | | | | |

capable of improving the baseline model on almost all corruption types, except for the "zoom blur". By looking into the bottom table where we compare to the AugMix method in detail, we can see that AdvBN can achieve comparable results with AugMix specifically on subcategories of "Weather" and "Noise", where it outperforms AugMix under more severe corruptions.

# E    MORE ABLATION STUDIES

We enclose more ablation studies in the appendix.

Table 10: **More ablation studies**.

| Model | ImageNet top1 acc. ↑ | ImageNet-C mCE. ↓ | ImageNet-Ins. top1 acc. ↑ | ImageNet-Styl. top1 acc. ↑ |
| --- | --- | --- | --- | --- |
| Standard | 76.1 | 76.7 | 67.2 | 7.4 |
| multiplicative $\delta$ | 76.5 | 68.6 | 71.1 | 14.1 |
| additive $\delta$ | 75.9 | 69.4 | 70.2 | 13.7 |
| $\gamma = 0.5$ | 76.5 | 69.4 | 70.8 | 14.4 |
| $\gamma = 1.5$ | 75.5 | 68.5 | 71.1 | 14.1 |
| $\gamma = 2.0$ | 75.3 | 68.5 | 71.1 | 14.0 |
| AdvBN* | 77.0 | 72.6 | 69.5 | 11.9 |

**Multiplicative vs. additive perturbations.**    We originally chose a multiplicative noise so that a single perturbation bound can be applied to various layers and architectures regardless of the range in which feature values lie. To adjust the perturbation bound for an additive noise for each batch of feature, we compute maximum mean and standard deviation values across channels, $\mu_{max}$ and $\sigma_{max}$. Then, in the projection step of the PGD attack, we project perturbations to the mean and standard deviation into the ranges $(-\epsilon \cdot \mu_{max}, \epsilon \cdot \mu_{max})$ and $(-\epsilon \cdot \sigma_{max}, \epsilon \cdot \sigma_{max})$, respectively. The model is denoted "additive $\delta$" in Table 10. This variant results in slightly degraded accuracy on each dataset, but notice that we use the same hyperparameter setting as the multiplicative case, and there are possible better settings for the additive case.

**Weighted adversarial loss**    Note that in Algorithm 1, our final objective is a equally weighted combination of clean loss and adversarial loss. It is possible that there are better way to combine the two term. In this section, we try adding a weigh to the adversarial loss, and see how the performance would be affected by it. That is, we now run the last step in Algorithm 1 by:

$$\theta \leftarrow \arg\min_{\theta} \left( \gamma \mathcal{L}(g_{\theta}^{l+1,L}(f_{adv}), y) + \mathcal{L}(g_{\theta}^{l+1,L}(f), y) \right) \tag{6}$$

We try $\gamma$ with range of values $[0.5, 1.5, 2.0]$ for this ablation study, adn the results are shown in Table 10.

**Standalone AdvBN**    In section 5.1, we propose to train AdvBN with a set of more sophisticated augmentation operations than standard ImageNet augmentation, which is proposed by auto augmentation(Cubuk et al., 2019) as the optimal set of data augmentation for the task of ImageNet classification. Therefore, the presented performance gain actually comes from both the image space data augmentation and AdvBN. The results prove that AdvBN, as a method orthogonal to image space data augmentation approaches, can be combined with and further improve these approached. In order to see the effectiveness of the standalone AdvBN method, we provide a results of training AdvBN with only basic ImageNet augmentations (random resized cropping and random flipping). The result for standalone AdvBN is denoted as "AdvBN*" in Table 10. We use the same hyperparameter setting for this result as we use for the "AdvBN + auto augmentation" experiment, and it is likely not the optimal setting.

