# OpenReview forum: "Prepare for the Worst: Generalizing across Domain Shifts with Adversarial Batch Normalization"
_ICLR.cc/2021/Conference — Reject_

### Official Review · AnonReviewer2 · 2020-10-27
**A straightforward approach**

**Rating:** 5
**Confidence:** 4

**Review:**

This paper proposes an algorithm for generalization to unseen domains. The algorithm performs adversarial training on the batch normalization coefficients. The authors provides experimental results showing the benefits of the proposed algorithm and also provides ablation study.

Strength:
I think the algorithm in this paper is simple and straightforward and the paper is easy to follow. The authors provided experiments on large scale dataset, some ImageNet-based datasets.

Weakness:
My major concern is that the experimental results does not fully validate the effectiveness of the proposed algorithm. For example, in Table 1, the results are very close to Augmix on Imagenet-C and Imagenet-Instagram. Only on Stylized-Imagenet, the proposed method shows benefits.
I have a hypothesis that the proposed AdvBN method may bias the model to be more robust to image style change (the images in Figure 2 and 3 kind of show this), but may not improve the robustness to other types of corruptions. Overall, it is not convincing that the proposed method can provide universal robustness gain to all kinds of corruption/domain changes.

Recommendations:
I would like to see more discussion on how the proposed algorithm affects different types of corruptions.  For example, there are 15 different types of image corruptions in Imagenet-C, and I suggest the authors to provide the test results on each type of corruption, so that the readers can better understand what types of corruption this method is more effective on.
If the authors cannot show that AdvBN provides universal robustness, I would like to suggest that the authors change the claims in the paper that the AdvBN improves the robustness to all the corruptions (or preparing for the worst), to a more gentle claim that the method is effective for some particular types of corruptions.
If possible, I would also like to see some experimental comparison between this method and distributional robust optimization, which I think is a more principled approach to getting general robustness. For example, can the authors compare with:
Volpi, et al, Generalizing to Unseen Domains via Adversarial Data Augmentation?


=======

After author response: I would like to thank the authors for providing the details of each corruption in ImageNet-C dataset. I understand that it might be hard to compare with Volpi et al due to lack of implementation details. However, I still feel that this submission is a bit lack of depth and thus it may not be a good contribution to the ICLR community. I would like to see more theoretical or experimental evidence that can help us get a deeper understanding of this approach. Overall I decided to keep my score.

---

> ### Author Response · Authors · 2020-11-23
> **Thank you for your feedback and recommendations.**
>
> We appreciate your constructive recommendations and we provide additional experimental results to support our method.
> As for your major concerns pointed out as the weakness of our work, our method indeed shows a larger margin on stylized images, but we wouldn't say a method is not effective because it has very close performance to state-of-the-art methods like AugMix on some task. We add detailed test results of AdvBN on the 15 categories of ImageNet-C corruptions in the appendix, and show that it provides improvements for 14 of the corruption types.
>
> As for the related work you refer to, we agree that it is a more principled approach. We try to compare to *Volpi, et al, Generalizing to Unseen Domains via Adversarial Data Augmentation* on their semantic segmentation results on SYNTHIA, because we want to see whether AdvBN can generalize well across different weather and lightning conditions. However, with limited implementation details given in that paper (e.g., how they split the training and testing data; what is the  learning rate, etc.) and no official segmentation code available, we are not able to reproduce the baseline model they use. With the same ResNet-50 + FCN architecture, our baseline model performs better than theirs by around 10% using the same training framework we use for Cityscapes. We understand that the comparison would thus be unfair, and we provide this result simply as an additional experiment to show the generalization ability of our method. We are working on comparing AdvBN with this method on classification tasks, and look forward to including it in the next version of this work.

---

### Official Review · AnonReviewer3 · 2020-10-28
**Official Blind Review #3**

**Rating:** 5
**Confidence:** 4

**Review:**

This paper proposes adversarial batch normalization (ABN) to perturb feature statistics. It is to makes CNNs more robust to image style or appearance changes. An ABN layer can plug into the middle of a CNN to finetune a pre-trained model. The CNN layers after the ABN layer can learn more robust representations from the perturbed features. It uses two groups of batch normalizations in the later layers: one for the adversarial features and the other for clean features. Experiments show it can improve robustness on image classification and segmentation.

Pros
1. Applying adversarial training to batch normalization statistics is well-motivated for image style robustness.
2. The method looks simple to implement.
3. Experiments show its effectiveness on two tasks: classification and segmentation.

Cons
1. Some typos exist in the paper, e.g., "additional compute" and "training times" in the last paragraph of Section 4.
2. The meaning of the last line in Table 2 is not clear. I have read the paragraph describing this ablation study (AutoAugment*) but still do not understand it. Does it mean ABN in Table 1 is with AutoAugment during the finetuning?
3. It is not clear whether ABN applies in conjunction with AugMix to further lower the corruption error. The current corruption error is close to AugMix.
4. Experiments only use one model, ResNet-50, which is insufficient. The layer ablation study does not explore the first network block and input. It is not clear whether the optimal layer changes if the network architecture changes.
5. The domain generalization setting from Cityscapes to GTA5 is unconvincing because Cityscapes are realistic data, while GTA5 consists of synthetic data. The opposite generalization makes more sense in practice.
6. The paper emphasizes the finetuning efficiency when compared with other methods trained from scratch. I think the pre-trained time also needs to be considered. For a new task, there is usually no well-known pre-trained models. This raises another question: how does the method perform to train a model from scratch?

In summary, the proposed ABN is simple and effective in improving model robustness to style changes. My main concern is that the experiments are insufficient and have some space for improvements. See points 2-6 in the above for details. Thus, I tend to rate it below the acceptance threshold.

Post-rebuttal updates

Thank the authors for the great efforts in addressing the concerns. The new experiments on two new backbones DenseNet-121 and EfficientNet-B0 show the method can work well with multiple backbones, which is good. However, my other concerns remain unsolved.

1. Combination with AugMix seems necessary to demonstrate state-of-the-art performance and its orthogonality.

2. I still think the generalization from GTA5 -> CityScape should be listed together with CityScape -> GTA5.

3. The running time comparison should take the model's pre-trained time into account in Table 8.

4. Regarding the blocks in the ablation study, I remember a ResNet50 for ImageNet has 4 blocks. Table 3 only lists 3 blocks (2,3,4). So the first block is not the first convolution layer.

Therefore, I still keep my original rating.

---

> ### Author Response · Authors · 2020-11-23
> **Thank you for your feedback and suggestions.**
>
> First we appreciate your careful reading, and we have fixed the typos in the updated version.
> As for your concerns:
> 1. *"Does it mean ABN in Table 1 is with AutoAugment during the finetuning?"*:
> Yes, we use a fixed set of augmentation operations that are found by the auto augmentation method. We agree that the total performance gain we show comes from both the image space data augmentation and AdvBN. We chose to present our results this way because we want to take advantage of another merit of our method, that it is orthogonal to image space data augmentation techniques, and that we can further improve these methods like what we are doing with auto augmentation. To address your possible further concern, we temporarily provide a plain AdvBN result (without Auto Augment) with the same hyperparameter settings we use for Auto Augmentation + AdvBN in appendix E, as we currently don’t have the time and resource to do a thorough search for hyperparameter of the plain AdvBN.
> 2. *"It is not clear whether ABN applies in conjunction with AugMix to further lower the corruption error. The current corruption error is close to AugMix."*:
> Thank you for the suggestion. We think combining AugMix and AdvBN is worth exploring. The fact that AdvBN can further improve auto augmentation gives us confidence that AdvBN can also work well with AugMix.
> 3. *"Experiments only use one model, ResNet-50, which is insufficient. "*:
> We now add two other models fine-tuned with AdvBN: DenseNet-121 and EfficientNet-B0, and prove that AdvBN can also improve the performance of these models. The results may be further improved, given that we don’t have time to do hyperparameter tuning for new architectures.
> 4. *"It is not clear whether the optimal layer changes if the network architecture changes."*:
> We explored different layers with EfficientNet-B0, and found the optimal layer of this architecture has one thing in common with ResNet-50, that they all output features with resolution (H/4 x W/4), where H and W are the height and width of the input image. However, this is not conclusive yet. We will include additional architectures and ablations in our final version.
> 5. *"The layer ablation study does not explore the first network block and input."*:
> We regard our method as a "feature space" adversarial training method. Input layer is not within our consideration.  Note that our method is motivated by an observation that the mean and variance of features maps encode style information, in which the feature extractor needs to have the ability to process the input to a certain level of abstraction. The first network block in ResNet-50, however,  is a single convolution layer, for which we assume it cannot work well as a feature extractor for AdvBN.
> 6. *"The domain generalization setting from Cityscapes to GTA5 is unconvincing"*:
> We agree that the opposite direction may be more useful in many applications.  However, our method works best for eliminating the reliance of models on fine textural features and color.  Thus, it makes sense to use this method trained on Cityscapes and evaluated on GTA5, where high frequency textural information has been removed. We have additionally added results of semantic segmentation on another dataset more suitable for our case, the SYNTHIA datasets, where we train models on one specific weather condition, and test on others. We demonstrate that AdvBN can constantly improve mIOU in this case by including detailed results in our updated paper.
> 7. *"For a new task, there are usually no well-known pre-trained models. This raises another question: how does the method perform to train a model from scratch?"*:
> We put forward our method as a simple and universal “additional step” to strengthen CNN models. Considering that AdvBN introduces noise to intermediate features, we think training with AdvBN can benefit from a relatively stable pre-trained model rather than training from scratch. Due to the limitation of time and resources, we have not obtained a converged model trained with AdvBN from scratch.  A practitioner can always approach a new task by pre-training in the standard fashion and then fine-tuning using AdvBN.

---

### Official Review · AnonReviewer1 · 2020-10-30
**A good new approach to generalization through normalization**

**Rating:** 6
**Confidence:** 4

**Review:**

The authors present an approach for tackling the generalization problem in neural network classifiers based on a new method of batch normalization.  The authors go on to apply this normalization in an autoencoder arrangement in order to show the relationship between the effect of the normalizer on the features and the image that they represent.  Finally, the method is evaluated by pre-training on imagenet, finetuning with the proposed batch normalization, and testing on three versions of imagenet with global transformations applied.

Pros:
Novel application of normalization idea
Clear presentation
Useful and relevant to the conference
Very good quantity and relevance of experimentation

Cons:
Would like to see more introspection on the results

All in all, I think this is a strong paper with only minor issues and would be a great addition to ICLR.  The idea presented is simple but also seemingly very strong in principle and I appreciate that it does not require much in the way of algorithmically complex calculations.  I am most happy about the presented level of experimentation which the authors have done a good job of a) contrasting to the state of the art, b) exploring their own model, and c) exploring other applications.  On that note, however, to explain more the “con” which I have listed, the main contribution here being the experimentation which proves the authors’ idea it would be very nice to have a discussion about what the results mean deeper than answering, “Does this prove the method?” either in the main paper or an appendix.  I have some examples listed below.  That to me would be something that could elevate the paper to the top 50% or more of papers.

Questions for the authors:
Why is AdvBN not improving on AugMix for imagenet-c but does for the other datasets does this indicate some drawbacks for the method?
Why does performance appear to get better in the ablation study on $\epsilon$ but then get worse after a certain threshold?

Post-rebuttal updates:
I thank the Authors for their response. After reading all the reviews and comments I feel that there are aspects of the proposed approach that are not fully understood, despite the improvements. For example, those related to AugMix, and providing fully symmetric comparisons between Cityscapes and GTA5, as several reviewers have pointed out. For these reasons, I have decided to revise my ratings as I also recognize the importance of these observations.

---

> ### Author Response · Authors · 2020-11-23
> **Thank you for your interest in our work.**
>
> Thank you for the suggestion, we agree and will include more introspection on the results in the future version of this work.
> Here are our response to your questions:
> 1. *"Why is AdvBN not improving on AugMix for imagenet-c but does for the other datasets does this indicate some drawbacks for the method?"*:
> The result of AdvBN we show is independent of AugMix. We have not explored using AdvBN to improve on AugMix, but we think it is worth trying in the future. As for our results, we find AdvBN achieves comparable results to AugMix on ImageNet-C and performs better on Instagram-ImageNet and Stylized-ImageNet. We think it is possible for a method like AdvBN that is designed for “unseen” domain generalization to not outperform some other methods on a specific dataset. In addition, we add a table in the appendix to show a detailed comparison to AugMix in terms of each severity degree on each corruption type of ImageNet-C.
> 2. *"Why does performance appear to get better in the ablation study on \epsilon but then get worse after a certain threshold?"*:
> We assume the magnitude of perturbation, which is controlled by \epsilon, can be “too small” or “too large”.  Too small perturbations may cause AdvBN to produce insufficient robustness, as the perturbed data distribution is nearly identical to the original.  Meanwhile, perturbations that are too large can result in features so corrupted that they are far beyond what is experienced at test time.  Like in standard adversarial training, there appears to be some compromise between robustness and accuracy.  A model trained with very large epsilon can be robust against extreme corruptions, but this robustness comes at the cost of lower accuracy on weaker corruptions and clean data.

---

### Official Review · AnonReviewer4 · 2020-11-01
**Choice of distribution shifts seems odd and sparse and performance improvements seem relatively small.**

**Rating:** 3
**Confidence:** 5

**Review:**

This paper introduces a method for improving robustness of neural networks to domain shifts by adversarially perturbing the feature statistics. This is a very interesting idea, by playing a middle ground between the worst case of PGD and not doing anything. My main problem about the paper is the evaluation and particularly the lack of evaluation of certain models and certain datasets.
Lets talk models first. It is unclear to me why the quite related AdvProp model is not evaluated here. Even if they are difficult to train the pre-trained models are available here: https://github.com/rwightman/pytorch-image-models. Same with the Noisy Student L2 model which doesn't have any sort of adversarially perturbation and performs much better on ImageNet-C than the best number reported here. For reference pretrained weights for both model types are available in the above link. With the availability of pretrained models it seems inexcusable to only have 5 arbitrary comparison points, especially when there are models with significantly better accuracy.

Furthermore, for stylized ImageNet and Instagram I would like to see what a resnet50 simply fine-tuned on those distributions look like.

Next on the distribution shift side, I'd also like to see more than just 3 distribution shifts. There have been two recent papers that do a metastudy of many distribution shifts: https://arxiv.org/abs/2007.08558 and https://arxiv.org/abs/2007.00644. A thorough evaluation on other distribution shifts can give a more complete picture of the advantage of the proposed approach to distribution shift rather than just 3 numbers out of context.

For these reasons I recommend rejection.

---

> ### Author Response · Authors · 2020-11-23
> **Thank you for the feedback.  We hope to clarify a few points.**
>
> 1. *"It is unclear to me why the quiet related AdvProp model is not evaluated here."*:
> The goal of AdvBN is very different from AdvProp.  The latter which uses adversarial examples as a data augmentation strategy to improve natural training. Instead, our approach aims to generalize to unseen domains during testing. Also, our proposed AdvBN does adversarial perturbation to the style/batch-norm values, which AdvProp does adversarial perturbations to image pixels.  Furthermore, in Table 1, all methods we compared to have released their official ResNet-50 models, but this is not the case with AdvProp.
> To address your concern, we add a separate table (Table 3 in the updated paper) solely to compare the results of our method with AdvProp on EfficientNet-B0 (Because it takes the least time to train, among all EfficientNet variants). We found that AdvBN outperforms AdvProp on ImageNet-Instagram and Stylized-ImageNet.
> 2. *"Same with the Noisy Student L2 model which doesn't have any sort of adversarially perturbation and performs much better on ImageNet-C than the best number reported here."*:
> We don’t think it is fair to compare the Noisy Student method with any of the methods in Table 1. All methods in Table 1 only use the original ImageNet images for training, which has approximately 1 million images. However, training a Noisy Student model requires access to 300 million additional images from an unpublished dataset.
> Also, we consider the Noisy Student method as a very different approach to solve the generalization problem, which is beyond the scope of data augmentation or adversarial training.
> Additionally, like the AdvProp method, the Noisy Student method has not yet provided official ResNet-50 models for a general comparison with other methods.
> 3. *"it seems inexcusable to only have 5 arbitrary comparison points, especially when there are models with significantly better accuracy."*:
> We find that the models you refer to as significantly better are actually obtained on significantly stronger baseline models. For example, a standard trained EfficientNet-B1 model (one of the baseline models of both methods you mentioned) already has better performance on ImageNet-C than that of “ResNet-50 + AugMix”.
> In addition, we wouldn't say our comparison points are arbitrary: we choose the most relevant methods to compare with, and we explain their relevance to our methods when introducing them in section 5.2.  We do this comparison to show that our improvement over baseline is non-trivial.
> 4. *"what a resnet50 simply fine-tuned on those distributions look like"*:
> In this paper, we are interested in improving universal generalization ability to "unseen" domains without access to their label information, which addresses the more realistic case where domain information is unknown at train time. Fine-tuning on a specific test distribution will undoubtedly achieve better results than any of the other methods considered in this paper - this is the equivalent of  “training on the test set” for domain generalization.
> Still, we see how one might want to know what is achievable by training on these sets. Our test result shows that a Stylized-ImageNet fine-tuned ResNet-50 can achieve 57.0% top-1 accuracy on the corresponding validation set. For ImageNet-Instagram, we have the result directly from *Wu, et al, Recognizing Instagram Filtered Images with Feature De-stylization*, that a ImageNet-Instagram fine-tuned ResNet-50 can have 74.5% top-1 accuracy on the corresponding validation set. For ImageNet-C, the ImageNet-testbed proposed in *Taori, et al, Measuring Robustness to Natural Distribution Shifts in Image Classification*, summarizes the performances of a broad range of models on a broad range of datasets, in which we can find how a ResNet-50 trained with a single corruption type from ImageNet-C performs on that corruption type. For example, a ResNet-50 trained with motion blur corruption can achieve Corrupted Error of 47.2 on the single motion blur corruption test data.
> 5. *"Next on the distribution shift side, I'd also like to see more than just 3 distribution shifts."*:
> We have presented our results on semantic segmentation in the original submission. Additionally, we now add results of semantic segmentation on a new dataset, SYNTHIA.

---

### Official Review · AnonReviewer5 · 2020-11-07
**Interesting idea of using BN for robust optimization, but the experimental part has many flaws**

**Rating:** 5
**Confidence:** 4

**Review:**

**Summary:**
The paper proposes a new adversarial training procedure that finds worst-case batch normalization (BN) parameters in some Linf-ball around the identity BN parameters. The authors show that this approach combined with AutoAugment significantly improves the accuracy on challenging datasets with domain shifts like ImageNet-C, Stylized Image-Net, and ImageNet-Instagram. Moreover, the authors show improved results on a semantic segmentation task. However, some experimental details presented in the paper require a further clarification.

**Pros:**
- The approach consistently improves on different datasets that measure robustness towards domain shifts (ImageNet-C, ImageNet Instagram, Stylized ImageNet).
- The approach doesn’t require any extra data (labeled or unlabeled).
- The approach also leads to better results for semantic segmentation.

**Cons:**
- Misleading presentation of the AdvBN results. In particular, in Table 1, “AdvBN” rather refers to “AdvBN + AutoAugment”. But it’s clear that adding AutoAugment to any other competing method would also improve them. Thus, one has to separately report results for “AdvBN (without AutoAugment)” and “AdvBN + AutoAugment” to allow a clearer comparison. Table 2 shows the results of *AutoAugment alone* (and apparently, there is a benefit of combining AutoAugment with AdvBN), but what one really needs to know is the performance of *AdvBN alone*.
- Comparison to Lp PGD training is incomplete. First of all, it’s not specified which model was used (L2- or Linf-trained and under which epsilon). Second, one should always do a grid search over the Lp-epsilon similarly as you did for the proposed AdvBN method in Table 2, and to report the best model. Otherwise it’s not even clear whether the proposed AdvBN method is better than standard Lp PGD training. Any Lp-robust models from Engstrom et al. (2019) is clearly suboptimal for the tasks considered in this paper since these models have much lower clean accuracy.
- I couldn’t find a discussion on this, so I assume you used AutoAugment with **all** its data augmentations including those that are present in ImageNet-C. If it’s true, then the comparison to AugMix is unfair as in their method they have removed all overlapping corruptions.
- This is a very important detail that should’ve been clearly discussed in the main part and not just in the appendix: *“The results we report in previous sections with regard to ImageNet, ImageNet-C and ImageNet-Instagram are obtained by using BN statistics corresponding to original features. We only use auxiliary BNs, which keep the batch statistics of adversarial features, to test performance on Stylized-ImageNet in Table 1.”*
And then I’m not sure what the results in Table 1 for AdvBN+AutoAugment tell us: that there exist 2 models obtained via AdvBN, and one of them is good on one domain shift, and another is good on another one? But how do you know at test time which of the 2 models to apply?
- The paper has multiple mistakes in the presentation of the proposed method:
    - Equation (4): maximization over the BN parameters is missing. It’s written that: *“This optimization problem contains a maximization problem inside the BNadv layer.”* But I don’t see how it can be true since the maximization should be done in front of the loss. Moreover, the expectation in Eq. (4) should be taken not with respect to pairs $(x, y) \sim D$, but rather with respect to batches $(x_i, y_i)_{i=1}^B$ to reflect the fact that adversarial BN introduces the dependency of the perturbation set on the sampled batch of points.
    - Algorithm 1 has multiple mistakes: (1) an additional loop over batches is missing, (2) not clear how $\delta_\mu$ and $\delta_\sigma$ are initialized, (3) *“Minimize the total loss w.r.t. network parameter”* -- argmin there seems to be inappropriate, I think what was rather meant is doing *one* step of gradient descent wrt $\theta$, (4) in the same place: there should be a clear distinction regarding when the loss is taken wrt a single data point, and when wrt a batch of points (this is particularly important since AdvBN introduces the dependency of the perturbation set on the batch).

**Suggestions:**
- First paragraph of Intro: what you refer to as “adversarial training” would be better to call specifically “Lp adversarial training”: *“While adversarial training makes networks robust to adversarial perturbations, it does not address other forms of brittleness that plague vision systems.”*
Because what you propose is also adversarial training but just with respect to a different perturbation set, and I assume you suggest that it does “address some other forms of brittleness”.
- *“Additionally, note that this module acts on a per-batch basis so that features corresponding to an individual image may be perturbed differently depending on the batch the image is in.”*
For me it sounds a bit strange that the perturbation set that you aim to be robust against depends on the current batch of images. I wonder if some batch-independent normalization schemes can be applied here with equal success?
- Equation (4): equal weights for both terms may be suboptimal. Thus, it would be good to include the weighting coefficient between the two terms of the objective in the ablation study.
- Table 2: the ablation regarding where to put the AdvBN layer is inconclusive since it had to be done with respect to different epsilons. It seems that the selected epsilon was just too high for conv3_4 and conv4_6 since the clean accuracy becomes worse than that of the standard model. The same also applies to “additive $\delta$”, it’s not clear what a single number tells us, there should be a grid search over the epsilon.

**Score:**
5/10. The paper proposes an interesting approach that can help to improve robustness towards domain shifts without requiring any extra data. I would be willing to increase the score if the paper improves its experimental part, in particular by properly reporting the results of AdvBN (see **Cons**) and its baselines.

---

> ### Author Response · Authors · 2020-11-23
> **We appreciate that you recognize the effectiveness and simplicity of our proposed method.**
>
> Thank you for your feedback and suggestions.
> As for your major concern, we agree that the total performance gain we show comes from both the image space data augmentation and AdvBN. We chose to present our results this way because we want to take advantage of another merit of our method, that it is orthogonal to image space data augmentation techniques, and that we can apply it on top of a range of methods including auto augmentation. To address your concern, we provide a plain AdvBN result with the same hyperparameter settings we use for Auto Augmentation + AdvBN in Appendix E, as we currently don’t have the time and resources to do a thorough search for hyperparameter for the plain AdvBN. We look forward to including properly reported results in the updated version of this work.
> As for you other concerns:
> 1. *"Comparison with Lp PGD"* :
> The fact that the Lp PGD method is not proposed to solve the domain shifts problem makes our comparison with PGD seem incomplete. We include it this way because we are trying to use official off-the-shelf models. To address this concern, we add another comparison with the AdvProp method, which is based on Lp PGD adversarial training but their models are tuned for better generalization cross domains.
> 2. *"Which of the 2 models to apply at test time"*:
> Instead of 2 separate models, models fine-tuned with AdvBN contain two sets of batch normalization layers inside this single model: “clean” BNs that tracks and stores statistics of “clean” feature, and “adversarial” BNs that tracks and stores statistics of features perturbed by AdvBN.  We choose to use one certain set of Batch normalizations before doing inference, which in the implementation level, is controlled by a parameter of the forward function. It is true that different BN statistics work well with different domain shifts, but we find that "clean" BNs are more potent in practice, given that we use it during inference for all domains in both classification and segmentation, except for the Stylized-ImageNet dataset.
> 3. *"the perturbation set that you aim to be robust against depends on the current batch of images"*:
> We agree that this dependence applies in the visualization scenario, where we only go through the dataset once. During training, however, samples are randomly shuffled for each epoch and models will be trained for many epochs, so the dependence on batch grouping is lessened. We do find your suggestion helpful, and we are interested in working with batch-independent normalization schemes.
> 4. *"equal weights for both terms may be suboptimal."*:
> We agree. We have done limited search for the weight settings and temporarily included it in the appendix E, as we need more thorough searches (and compute time) to reach conclusions.  Computations are ongoing.
> 5. *"The ablation is inconclusive"*:
> In the ablation study, we are trying to show how single factors affect the performance respectively, so we keep the epsilon to be the same when showing the results. We have tried other epsilon values for the case of conv3_4 and conv4_6, and we observed that the differences are relatively small and all results for these layers are not as good as conv2_3.

---

> > ### Comment · AnonReviewer5 · 2020-11-24
> > **Comparison with AdvProp is helpful, but the concern with the auxiliary vs standard BN remains**
> >
> > Thanks for the detailed answer, extra experiments, and improving the typos/mistakes in the text.
> >
> > 1. *"To address this concern, we add another comparison with the AdvProp method, which is based on Lp PGD adversarial training but their models are tuned for better generalization cross domains."*
> >
> > Thanks for adding this comparison. As far as I understand, AdvProp also used AutoAugment during training, so this comparison sounds fair. Although it's unclear which method performs better since the improvement on Imagenet-Instagram and ImageNet-Stylized over AdvProp seem to be relatively small (+0.7% and +1.1%), while on ImageNet-C the performance is worse (-2.5%).
> >
> > -----
> >
> > 2. *"It is true that different BN statistics work well with different domain shifts, but we find that "clean" BNs are more potent in practice, given that we use it during inference for all domains in both classification and segmentation, except for the Stylized-ImageNet dataset."*
> >
> > I still don't see how this approach can be meaningful. I see that *"we choose to use one certain set of Batch normalizations before doing inference"*, but then it's precisely the problem -- you assume you know the domain shift and you use a different model for it. But how do you know which domain shift are you going to encounter on your next image? **I would be very curious to know the performance of AdvBN on ImageNet-Stylized using the standard BN instead of the auxiliary one.** This part seems critical to me.
> >
> > -----
> >
> > 3. Thanks for clarifying this. It's not a major concern for me, but just something that conceptually complicates the method.
> >
> > ----
> >
> > 4. Good to see that, I think providing a full ablation study for gamma would definitely be helpful.
> >
> > ----
> >
> > So far I keep my current score 5/10 but I would be willing to increase it if the authors provide favorable results of AdvBN on ImageNet-Stylized using the *standard BN* instead of the auxiliary one (I know that there is not so much time left until the end of the rebuttal, but the authors' response came quite late; and hopefully this evaluation can be done relatively fast as it doesn't require retraining a model, just plugging in a different set of BN parameters). Moreover, the comparison should include both ResNet-50 (to compare to other entries in Table 1) and EfficientNet-B0 (to compare to AdvProp with the same network).
> >
> > For me, the main problem currently is that because of the auxiliary BN issue, it's still unclear whether the proposed method is more effective than simple Linf adversarial training used in AdvProp.

---

> > > ### Author Response · Authors · 2020-11-24
> > > **Thank you for the reply. We provide you with the additional results.**
> > >
> > > Thank you for your interest in our results. We tested our models on ImageNet-Stylized dataset and found that, for ResNet 50, the top-1 accuracy using *standard BN* is 9.7%, and for EfficientNet-B0 this accuracy is 13.5%. We expect this performance to drop, because we assume ImageNet-Stylized dataset has extremely different distribution from others. Compared to other models in Table 1, we assume our model rely more on the normalization layers (with proper BN statistics) to achieve better performance on ImageNet-Stylized because model parameters have only been trained with augmented features for 20 epochs, while for other methods, for example AugMix, the model is trained with augmented data for 180 epochs. We assume good normalizations can help ease the distributional difference between datasets and this benefit can be more obvious on Stylized ImageNet with our model, where the data is extremely far away from the standard data distribution, while our model are relatively close to a standard trained model.
> > > In addition, we want to mention that our segmentation results in Table 5 and Table 6 are obtained without auto augmentation in the input space, and only use the *standard BN* during inference. It shows that the *standard BN* can generalize well to various domain shifts that we consider practical while not as extreme as Stylized ImageNet.
> > > As for the comparison to AdvProp, we expect our model to achieve better results if we fine-tune the EfficientNet model for a number of epochs that is proportional to its training scheme, which trains the model from scratch for over 300 epochs. (This experiment is ongoing.)  In other words, the model we use for comparing to AdvProp in Table 3, is only fine-tuned for 20 epochs, while an AdvProp model is trained for over 300 epochs.

---

> > > > ### Comment · AnonReviewer5 · 2020-11-24
> > > > **Overall comparison with AdvProp is not in favor of AdvBN (although there are differences in the number of epochs)**
> > > >
> > > > I see that it's hard to make a proper comparison with the original AdvProp given limited computational resources and time. Perhaps, another option is to fine-tune AdvProp from a standardly trained model also for 20 epochs similarly to what you do with your method for a fair comparison (with a grid search over the Linf epsilon).
> > > >
> > > > To conclude, despite the proposed approach is conceptually interesting, I think there is not enough evidence to say that it outperforms simple Lp adversarial training which is a natural baseline if one sticks to a single BatchNorm at test time. The results on segmentation in Table 5 and 6 are promising, but it could be that a similar gain potentially can be achieved also with Lp adversarial training. This makes me keep the original score 5/10.

---

> > > > > ### Author Response · Authors · 2020-11-25
> > > > > **Results of segmentation using AdvProp**
> > > > >
> > > > > Thank you for the suggestion. We understand your concern, and see possible way to improve our experiment section.
> > > > > We would like to compare to AdvProp in the fine-tuning scenario, and given more time, we will include it in the next version of this work. Also, we think it is a good point that you mentioned *"The results on segmentation in Table 5 and 6 are promising, but it could be that a similar gain potentially can be achieved also with Lp adversarial training."*, and we are also interested to see whether this is the case.
> > > > > Given pre-trained baseline models on hand, we are able to fine-tune a segmentation model on one of the Synthia dataset with the AdvProp method, using same amount of epochs as for AdvBN. Given limited time we have, we ran one setting using the epsilon value that is supposed to be optimal for EfficientNet-B0 found by authors of the AdvProp in Table 2 of Xie, et al, *Adversarial Examples Improve Image Recognition*.
> > > > > We summarize the results below:
> > > > >
> > > > > |          |      |      | Highway |        |        |
> > > > > |:--------:|------|:----:|:---------:|:------:|:------:|
> > > > > |          | Dawn |  Fog | Night   | Spring | Winter |
> > > > > | baseline | 18.6 | 21.0 | 16.9    |  21.6  |  15.3  |
> > > > > |  AdvProp | 12.1 | 19.7 | 10.5    |  21.7  |  18.4  |
> > > > > |   AdvBN  | 21.6 | 24.2 | 22.2    |  27.2  |  19.8  |
> > > > >
> > > > > This table corresponds to the right-hand side table in Table 6. All three models are trained/fine-tuned on the N.Y. Like C./Spring dataset, and the metric we use here is mean IOU.
> > > > > We believe our method is well motivated, that perturbing BN statistics would be a fast and straightforward way to improve generalization ability.

---

### Author Response · Authors · 2020-11-23
**Paper Revision: Added ImageNet results on more network architectures; added semantic segmentation result on new dataset.**

Firstly, we want to thank all reviewers for providing feedback and constructive suggestions for improving the experiment section. We’ve updated our paper by providing more experimental results:
* Applied AdvBN to DenseNet-121 and EfficientNet-B0 for ImageNet classification.
* Evaluated AdvBN on traffic scene semantic segmentation across different weather/illumination/season conditions on the Synthia dataset.
* Compared with AdvProp on EfficientNet-B0.
* Provided detailed performance on ImageNet-C in Appendix D by including corrupted error on each severity degree of each corruption type.

Note that our results for DenseNet and EfficientNet may be updated later given more time for tuning hyperparameters. We also look forward to including more architectures.
Second, we thank the reviewers for carefully reading our work, pointing out typos, and presenting possible ways to improve notation. We have changed notation in Eq(4) by adding the maximization, and we fixed the typo in the condition of the outer loop of the algorithm, as well as typos in Section 4.
Additionally, in order to meet the page limits, we move the ablation study on “additive noise” to the appendix.

---

### Decision · Program_Chairs · 2021-01-07
**Final Decision**

**Decision:**

Reject

**Comment:**

This work proposes a novel, interesting and simple technique to improve the model robustness to distribution shift. The proposed method is called Adversarial Batch Normalization (AdvBN) which is based on adversarial perturbation of BN statistics. Authors provide extensive experiments to show the effectiveness of AdvBN. All reviewers agree that the proposed method is interesting and novel. The main concern of reviewers is about the some of the details of the empirical evaluation of the proposed methods which makes its effectiveness less clear. In particular, the following concerns are shared among the reviewers:

1- Authors give different treatments to Stylized-ImageNet compare to other tasks by using auxiliary BN at inference time instead of standard BN and further results provided by authors show that the improvement over previous methods disappear if they use standard BN for inference on Stylized-ImageNet. I think authors could mitigate this issue by further investigation or providing a better explanation on why they have a different treatment for Stylized-ImageNet (other than the fact that auxiliary BN has a better performance on that task). The other potential remedy is to come up with an automatic way to decide which one to use at the inference time using a batch of "unlabeled" validation data.

2- The improvement of AdvBN over AugMix and AdvProp (which was added during the rebuttal) is not clear. In particular, both methods improve over AdvBN on ImageNet-C. If standard BN is used for AdvBN on Stylized-ImageNet, then both AugMix and AdvProp improve over AdvBN. That only leaves ImageNet-Instagram as an ImageNet variant where AdvBN shows a clear improvement over AdvBN and AugMix. A potential solution is to try combining AugMix and AdvBN (not sure if AdvProp could be combined effectively) to see if there is a way to get maximum benefit out of these methods.

3- The empirical section could be improved by doing experiments in a systematic way. That is for any choices made in the experiment design, there should be a reason that is explained clearly. For example: 1) applying the same type of data-augmentation on all methods (or reporting all methods with and without data-augmentation). 2) compare to all baselines on ResNet-50 and then pick the top 2 baselines (say AugMix and AdvProp) and then compare them on DenseNet and EfficientNet. 3) comparing with the same baselines as (2) on the segmentation task.

Finally, I want to thank authors for engaging with reviewers, running many experiments during the rebuttal period and updating the paper accordingly. I also want to reassure authors that my final evaluation of the paper is based on: 1) reading all reviews and responses 2) weighing the reviews based on their substance, quality and engagement of reviewers 3) looking at the initial and final revision of the paper. In particular, even though the average score of this paper is low, in my opinion it is a borderline paper. After taking all of the above into account, my decision is to recommend rejection. Even though the proposed method is very interesting, there are three clear valid concerns all of which can be addressed as I suggested above. Without addressing those concerns, the empirical advantage of the proposed method is not demonstrated properly. I think after addressing those concerns this paper will be in a much better shape, more useful for ML community and hence receives the attention it deserves. I sympathize with authors that their efforts during the rebuttal period did not result in improving reviewers' scores but I want to emphasize that I did take all those updates into account when making my recommendation for this paper.